# HIV in the Brain: Identifying Viral Reservoirs and Addressing the Challenges of an HIV Cure

**DOI:** 10.3390/vaccines9080867

**Published:** 2021-08-05

**Authors:** Michelle K. Ash, Lena Al-Harthi, Jeffrey R. Schneider

**Affiliations:** Department of Microbial Pathogens and Immunity, Rush University Medical Center, Chicago, IL 60612, USA; michelle_k_ash@rush.edu (M.K.A.); lena_al-harthi@rush.edu (L.A.-H.)

**Keywords:** HIV, CNS, reservoir, cure, astrocytes, microglia, inflammation

## Abstract

Advances in antiretroviral therapy have prolonged the life of people living with HIV and diminished the level of virus in these individuals. Yet, HIV quickly rebounds after disruption and/or cessation of treatment due to significant cellular and anatomical reservoirs for HIV, which underscores the challenge for HIV cure strategies. The central nervous system (CNS), in particular, is seeded with HIV within 1–2 weeks of infection and is a reservoir for HIV. In this review, we address the paradigm of HIV reservoirs in the CNS and the relevant cell types, including astrocytes and microglia, that have been shown to harbor viral infection even with antiretroviral treatment. In particular, we focus on developmental aspects of astrocytes and microglia that lead to their susceptibility to infection, and how HIV infection propagates among these cells. We also address challenges of measuring the HIV latent reservoir, advances in viral detection assays, and how curative strategies have evolved in regard to the CNS reservoir. Current curative strategies still require optimization to reduce or eliminate the HIV CNS reservoir, and may also contribute to levels of neuroinflammation that lead to cognitive decline. With this in mind, the latent HIV reservoir in the brain should remain a prominent focus when assessing treatment options and overall viral burden in the clinic, especially in the context of HIV-associated neurocognitive disorders (HAND).

## 1. Introduction

Since its emergence as a human retrovirus in the 1980s, HIV has existed as a global pandemic for nearly 40 years. While great advances have been made in treatments for people living with HIV (PLWH), a cure still remains elusive to researchers and clinicians alike. This is in part due to the latent HIV reservoir, defined as stable accumulations of replication-competent forms of virus in a myriad of anatomical sites throughout the body, that are still being defined [1]. Ho and colleagues recently re-evaluated the size of the HIV reservoir by performing viral outgrowth assays to assess intact genomes of proviruses and identified a large pool of non-induced proviruses that are activated in a random manner, noting that if these proviruses can be induced in vivo, the latent reservoir may be much larger than historically determined [2]. Certain anatomical locations have been shown to harbor latent HIV infection, including the gut [3] and the brain [1]. It has been suggested that the persistence of HIV within the gut may be due to genetic diversity within different gut tissue compartments (e.g., esophagus, duodenum, colon, etc.) allowing the virus to evade antiretroviral therapy mechanisms [4]. Similarly, within the CNS, HIV compartmentalization within the brain allows for evolution of virus that can evade clearance mechanisms and exhibit a unique phenotype that only permits viral replication contingent upon immunosuppression [5].

The brain is not immune to viral infections. In addition to HIV, a number of viruses can invade the brain including cytomegaloviruses (CMV), John Cunningham virus (JCV), and Herpes simplex virus (HCV) [6,7]. HIV, in particular, invades the brain within 1–2 weeks of infection based on human and non-human primate studies [8,9]. Within the first 100 days of HIV infection, magnetic resonance imaging (MRI) scans reveal brain stem and third ventricle expansion, as well as a loss of white matter integrity and mild neurocognitive deficits, meaning that the alteration of brain tissue within the first several months of HIV infection is probable and advocates for early treatment [10]. In fact, it was recently shown that individuals that start early cART treatment, median 17 days after infection, have normal levels of inflammation/immune activation markers in the CSF 96 weeks later [11]. In some individuals, HIV RNA has been detected within the cerebrospinal fluid just eight days after exposure to virus [9]. In the brain, the virus sets the stage for inflammatory responses. While prior to combined antiretroviral therapy (cART), HIV caused dementia in 25% of PLWH, in the cART era that number has declined to less than 5% and is mostly due to lack of antiretroviral access or issues with drug compliance. Nonetheless, HIV in the era of cART causes a spectrum of neurologic disease termed HIV associated neurocognitive disorders (HAND). What drives HAND is not entirely clear and it may be driven by low-level virus replication in infiltrating immune cells and/or resident brain cells as well persistent inflammatory responses in the absence of virus replication. The utilization of ART technology and its refinement over the past 35–40 years has decreased the incidence of severe HAND, to milder/moderate symptoms [12], although these symptoms impact the daily lives of PLWH experiencing neurocognitive impairment. Even with the advances made with cART over the last two decades, the incidence of HAND remains in the range of 20–50% for PLWH [13].

### Cellular HIV Reservoirs

HIV neuroinvasion occurs both dependently and independently of a break down in the blood brain barrier (BBB) (Figure 1). Classically, the trojan horse model is often used to describe HIV neuroinvasion through the trafficking of HIV infected lymphocytes (CD4+ T cells and monocytes) into the brain (Figure 1A). Once in the CNS, HIV can then infect glial cells (microglia and astrocytes) through direct infection mediated by CD4/chemokine fusion of HIV gp120 in the case of microglia or through endocytosis for astrocytes (Figure 1D). Lymphocytes can also enter the brain through the meningeal lymphatic system and activate inflammatory pathways to permeabilize the vessels and cross the vessel walls into the CNS (Figure 1E). Reservoirs, including microglia, macrophages, and astrocytes, promote a prolonged inflammatory state (Figure 1A–C) which can ultimately lead to neuronal dysregulation and/or cell death (Figure 1C), which plays a role in cognitive decline.

## 2. Microglia/Macrophages

Microglia, the resident macrophages of the CNS, are established during embryogenesis from progenitor cells in the fetal yolk sac, although there is also evidence of an additional pathway in which blood monocytes invade the CNS and then differentiate into microglial cells [14,15]. Several hypotheses exist about the differentiation of microglia, but most point to an activation signal coming from damaged or enflamed CNS tissue, typically through the TGF-β signaling pathway [16]. Others indicate that microglial differentiation can only begin after CNS pre-conditioning, which disrupts the blood brain barrier to recruit monocytes into the brain [14]. Monocytes, on the other hand, also serve as precursors for brain macrophages after being trafficked across the BBB [16]. Monocyte-derived macrophages are much more susceptible to HIV infection, while peripheral blood monocytes have protective anti-HIV mechanisms, indicating that macrophages seeded in the brain are one of the major carriers of HIV infection [17]. Brain macrophages that are compartmentalized as latently infected reservoirs can then re-seed competent infection after ART suppression [18].

Walsh and colleagues (2014) have shown that HIV infected microglia produce higher levels of interleukin (IL) IL-1β, IL-18, and caspase-1, the three inflammatory molecules comprising the NLR family pyrin domain containing 3 (NLRP3) inflammasome [19]. These findings were recapitulated in feline macrophages infected with feline immunodeficiency virus (FIV). Additionally in an in vivo model FIV induced neuronal loss and neurocognitive deficits [19]. This aligns with an earlier paradigm establishing that HIV-infected macrophages in the CNS can release neurotoxic proteins, leading to neuroinflammation and damage, as well as increasing the susceptibility of microglia for HIV infection [7]. Infection-susceptible macrophages and microglia migrate to areas of infection, then release inflammation-inducing superoxide as a consequence of the Negative Regulatory Factor (Nef) protein from HIV [20]. Additionally, the β-catenin signaling pathway is integral for a number of functions within the CNS, including the release of neurotransmitters, increasing synaptic strength, and memory consolidation [21]. Monocytes migrating into the CNS to differentiate into macrophages experience a downregulation of β-catenin, leaving them vulnerable to HIV infection [22].

Microglia have been shown to be susceptible to HIV infection, and more importantly, go on to harbor latent virus and evade apoptosis, serving as a reservoir site in the CNS [23]. Earlier models of HIV infection in microglia indicated a higher presence of HIV+ microglia in the CNS, but most of these studies utilized human post-mortem tissue of donors with severe neurocognitive deficits [24]. Updating these models and utilizing techniques with higher sensitivity is integral to understand the presence of the latent HIV reservoir within the CNS.

As HIV invades the brain, a powerful inflammatory response is activated that can cause neurological deficits. The presence of leukotriene C_4_ has been shown to promote migration of CD4^+^ T cells across the blood-brain barrier, which then enhances transfer of HIV infection to macrophages/microglia in the brain [25]. When this occurs, GABAergic and dopaminergic neurons act to silence HIV expression in microglia and brain macrophages and promote the induction of latency in these cells [26]. Therefore, microglia and brain macrophages are critical targets of the HIV latent reservoir and must be included in future cure strategy approaches.

## 3. Astrocytes

Astrocytes are a significant cell population in the brain and have a variety of functions including secretion of synaptogenic molecules, synaptic pruning, maintenance of CNS homeostasis, regulation of blood flow, and responding to injury and disease [27]. Astrocytes are a diverse and heterogeneous population, including various phenotypes based on CNS region: protoplasmic, fibrous, and radial astrocytes [28]. There are nine known astrocyte phenotypes: tanycytes, radial, Bergmann glia, protoplasmic, fibrous, velate, marginal glia, perivascular, and ependymal glia [29]. Infectivity varies based on brain region, with higher anti-HIV responses in the spinal cord and cortex than in the cerebellum [30]. Astrocytes are targeted by a number of viruses, including murine leukemia virus, feline immunodeficiency virus, human T Cell tropic virus type 1, and simian immunodeficiency virus, and due to the overall abundance and dynamic nature of astrocyte phenotypes based on location in the brain, astrocytes represent an important potential HIV CNS reservoir [31].

Astrocytes have been shown to be HIV infected in vitro by several groups and in vivo through the use of in situ hybridization and laser capture dissection followed by nested PCR [32,33,34,35,36,37]. More recently, the integration of HIV into astrocytes has been demonstrated, albeit at a low percentage. However, when you consider the size of the total astrocyte population this is still a potentially important reservoir [38]. Using human post-mortem brain tissue from donors under cART and through using RNA and DNA scope along with laser capture microdissection to pull infected astrocytes, a recent study revealed between 0.1–1% of human astrocytes harbor integrated HIV DNA [39]. This low rate of infection can disrupt the blood brain barrier through affecting gap junction integrity as well as uninfected bystander effects [40]. In another study, HIV infected astrocytes disrupted beta-catenin signaling, leading to metabolic dysregulation, increased inflammation, and neuronal injury [41,42]. Building upon these findings, work from the Brack-Werner lab has further established the astrocytes are part of the latent reservoir through experiments using latency reversing agents (LRAs) in vitro [43,44]. Therefore, although astrocytes may not be the dominant cell type in the HIV latent reservoir, their contribution should be considered when developing curative strategies.

Newer technology using clinically relevant in vivo data exists, allowing the field of HIV/astrocyte research to investigate the dynamics of cellular viral infection in-depth. Models of astrocyte infection of HIV have been established, including the use of the multipotent human neural stem cell line HNSC.100, in which astrocytes can be differentiated and productively infected with HIV long-term [45]. The co-localization of CD81 on astrocytes with HIV acts as an additional protective compartmentalization mechanism, allowing cells to transfer HIV infection to T cells which can then migrate out of the CNS and into the periphery [46]. This model has been reinforced through identification of CNS viral populations in peripheral tissues re-seeded with T-cell tropic viruses [47]. A novel chimeric model of human astrocyte/human peripheral blood mononuclear cells in NOD/*scid*-IL-2Rgc *null* (NSG) mice (huAstro/HuPBMC mice) recently demonstrated that astrocytes are a reservoir for HIV and support HIV egress from the brain to peripheral organ through lymphocyte trafficking [47]. HIV egress from the brain (originating form HIV infected astrocytes) to periphery also occurred under cART in these animals. This finding highlights the potential role of the brain as source of rebound virus under maximal viral suppression. PLWH experience viral blips even under cART or virus reactivation after cessation of cART. Although the primary target of HIV latency are the resting CD4+ T cells, during viral blips or cessation of cART the sequence of the virus that emerges is different than virus in resting CD4+ T cell suggesting that there are distinct anatomical sites and cellular sources for latent HIV [48,49]. The brain could be one such site, as demonstrated in these humanized animals. Another new model shows astrocyte infection in a CXCR4-dependent manner, where immature HIV particles can be transferred from trafficked lymphocytes to bind CXCR4 on astrocytes and promote cell fusion [50].

Although astrocytes do not have CD4 and co-receptors for a classical receptor-mediated viral entry, they have been shown to acquire HIV infection via pH-dependent endocytosis, which typically eliminates the majority of viral particles, but could be a significant pathway for HIV to establish latency within astrocytes [51]. There is evidence for astrocytes to harbor productive infection of HIV through direct contact with CD4^+^ infected T lymphocytes, although at low levels of detectability [52]. While this novel concept makes the case for astrocytes having a different route of infection, it also challenges the idea of the CNS being an immune-privileged area, a concept that has evolved over decades of CNS research to include the newly rediscovered lymphatic system within the brain [53]. Traditionally, viral particles would enter the CNS and cause infection, however there are also other ways that HIV can enter the CNS without active viral particle trafficking across the blood brain barrier. The meningeal lymphatic system serves as a source of HIV+ cells that circulate into the CNS from peripheral lymph nodes, introducing the lymphatic system as a key player in HIV migration [54]. Utilizing the newly re-discovered lymphatic pathway into the CNS can be a powerful tool in identifying and eliminating CNS reservoir entry sites.

## 4. Outstanding Questions & Direction of the Field

Compounding evidence indicates that microglia and astrocytes serve as reservoirs for HIV in the CNS. However, there are still outstanding questions regarding the evolution of these cells as reservoirs, and how they can promote re-infection in peripheral sites. For astrocytes specifically, while re-infection has been modeled in vitro, as well as in vivo in mice, a higher animal model utilizing animals such as a non-human primates (NHP) would reinforce astrocytes’ ability to infect circulating T cells that can re-seed infection at peripheral sites. Additionally, this would highlight trafficking patterns to target in humans for new advancements in cART treatment. However, the NHP model is not without its challenges. Infection of NHPs with SIV, the simian form of HIV, can lead to a variety of differences from human disease progression, depending on species. Comparing Chinese versus Indian rhesus macaques, Chinese rhesus macaques will exhibit slower disease progression when challenged with SIV, while Indian rhesus macaques, virus is more fit for replication and initiates faster depletion of CD4^+^ T cells [55]. Models using pig-tailed macaques have shown viral persistence in brain macrophages, but relied on virus tropic for both CD4^+^ T cells and macrophages, making the study indirectly comparable to other models with restricted phenotypes of virus [18]. Because of these pitfalls, some SIV models to date do not accurately represent disease progression and may not reflect viral loads or sanctuary sites in the CNS and call for revision of the model. Regardless, these models are the closest available physiological system compared to humans, and therefore are critical to our understanding of HIV manifestations in the brain.

In addition to strengthening NHP models of HIV infection, utilization of advanced neuroimaging techniques can assist in reservoir research. RNA in-situ hybridization has become a powerful tool to evaluate viral loads within the CNS, and has detected replication-competent cells following treatment with latency reversal agents [56]. Higher-sensitivity detection tools including surface-enhanced Raman scattering (SERS) spectroscopy [57] and semi-nested qPCR [58] on both patient and animal model tissue lead to data that is more reflective of reservoir size and viral load as compared to previous post-mortem human or animal model necropsy tissue. The utilization of more sensitive technologies, especially in combination with more competent models, going forward will better serve the HIV research field in determining sanctuary sites as well as providing specific anatomical and cellular targets for drug delivery in patients.

### HIV Cure Strategies with Regard to CNS Reservoirs

As evident from the above sections, the existence of HIV reservoirs serves as the main obstacle to finding an HIV cure. The current standard of care for PLWH is cART, a combination of medications that can effectively suppress viral loads within the blood. However, cART is not a “catch all” for HIV infection. Once a patient stops cART treatment, viral loads will increase to be detected in the blood once again, proving the therapy’s limitations as a cure for HIV due to its inability to effectively target latently infected cells [59]. SIV-infected macaques have HIV RNA present in brain tissue and these levels are comparable within cART-treated animals, indicating that the brain may be a sanctuary for HIV regardless of cART status [60]. It has also been shown that, after long-term cART treatment and viral suppression, viral genomes persist within the CNS [61]. cART treatment can suppress HIV infection in the blood, but drug penetration levels within the brain remain less than ideal, and allow for continuity of viral presence within the brain [62]. Through this research, it is evident that while cART has greatly improved prognoses for PLWH, the treatment must evolve to include more potent penetration of the CNS. Without a cure, many HIV-associated complications will continue to affect patients even in the era of cART.

A popular strategy toward a cure for HIV is known as “block and lock” (Figure 2A). This treatment disrupts HIV transcription with a variety of small molecule compounds, and is incredibly scalable compared to expensive and time-consuming treatments such as stem cell transplants [63]. Tat inhibitor didehydro-Cortistatin A has been shown to induce epigenetic silencing of HIV and delay viral rebound up to 19 days in ART-treated mice, making it a promising new drug for use in tandem with cART [64]. RNA-induced epigenetic silencing is also a promising treatment, using short hairpin RNAs (shRNAs) to block promoters on HIV provirus genomes, and may be able to activate “super latency” in which chromatin architecture is altered to inhibit removal of shRNAs that would allow for transcription of proviral genes [65]. Block and lock strategies are powerful tools for inducing HIV latency, especially due to their low cost and ease of delivery, but they do not eliminate viral presence in cells. This is a potential issue for CNS reservoirs, as latent virus can still cause prolonged low-level inflammation that is detrimental to brain tissue [66].

The most drastic yet distinct anti-HIV therapy to date consists of latency reversal agents to combat reservoir persistence, also known as “shock and kill”. Latency reversal agents are drugs delivered to patients meant to induce replication of virus from reservoir cells, allowing for target and elimination of these cells [67]. The reactivation of latent cells would be an effective way to clear reservoirs from patients, but it comes with the caveat of potential toxicity, especially in organ systems like the CNS. Patients who already show neurocognitive decline may undergo further degradation if latency reversal agents are used and indirectly activate a massive inflammatory response against re-activated reservoir cells in the CNS. Latency reversal agents can also cause inhibition of CD8^+^ T cells, a dangerous action in already CD4^+^ T cell deficient PLWH [68]. Not only does this pose a potentially side effect of latency reversal agents, but it also highlights a gap in latency reversal knowledge.

An additional strategy for blocking HIV infection in the brain involves genetically modifying cells. HIV reservoir research has evolved to include methods such as delivery of guide RNAs to cells harboring proviruses through the CRISPR/Cas9 system, which have been shown to reduce instances of reactivation of latently infected astrocytes [69]. The system has been elaborated by the Brack-Werner lab, showing the delivery of Adeno-associated virus-based vectors to astrocytes via CRISPR/Cas9 is effective at reducing provirus reactivation [70]. While efforts have been made to apply these techniques in vivo, the combination LASER ART and CRISPR/Cas9 treatment eliminated HIV presence in only two of six mice, and had minimal effects in the brain [71]. It is important to note that this technology is relatively new and requires further investigation to be considered as an effective HIV curative strategy, but should be modified to have more potent CNS penetration while preserving the brain’s natural homeostasis.

One of the main challenges with cART is that it has low penetration index into the CNS [72]. Will either of the aforementioned cure strategies cause a similar issue? A number of latency reversal agents to stimulate viral replication have been studied, but there is little research detailing how well drugs targeting the reactivated cells can cross the blood-brain barrier and eliminate CNS reservoirs.

## 5. Conclusions

HIV has existed as a global health crisis since its emergence in the 1980s and still eludes a cure. Only two individuals have been effectively cured of HIV to date, one of which, the Berlin Patient, recently passed away due to complications from leukemia [73,74,75]. Reservoirs in the central nervous system provide evasive routes for viral DNA to lay dormant without activating immune clearance mechanisms, and also contribute to a prolonged neuroinflammatory state leading to neuronal apoptosis and cognitive dysfunction. New insights on cellular trafficking into the CNS show potential to block back door viral entry into the CNS, while older and current HIV therapies need to be revisited to determine their toxicity and effectiveness in penetrating the CNS. This review addressed the current state of the cellular CNS reservoir field and how the current paradigm of HIV treatment can affect the CNS reservoir. Going forward, these results indicate the possible need for new or updated HIV treatments to address the cellular reservoir pool, although not just in the CNS. The identification of reservoirs and therapies to eliminate them, as well as the adverse pathological events they lead to, is paramount for the future of HIV patient care and improving quality of life.

## Figures and Tables

**Figure 1 vaccines-09-00867-f001:**
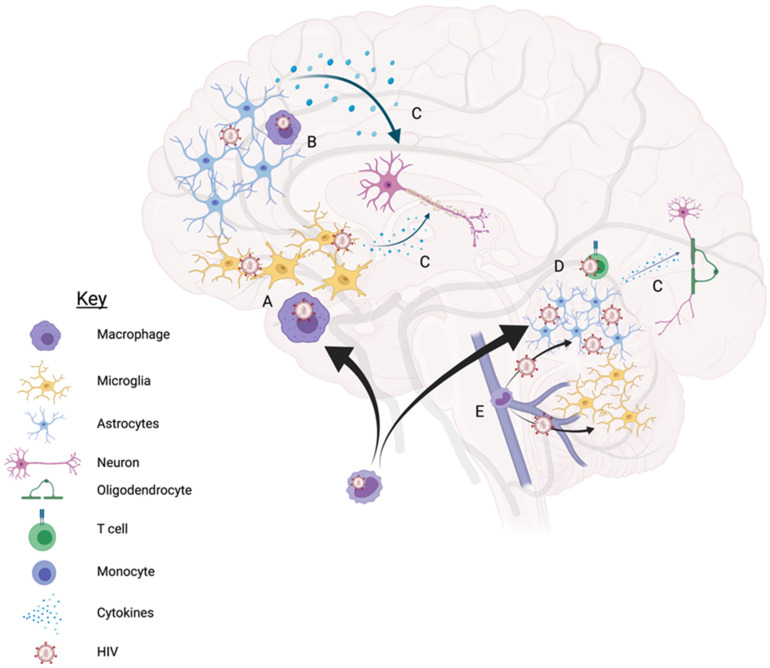
Depiction of cells in the brain and their interactions with HIV. (**A**) HIV+ macrophages (purple) circulate through the blood brain barrier and pass through into the CNS, where they can release viral particles and infect microglia (yellow). (**B**) direct contact with infected macrophages infects astrocytes (orange), which have the capability to pass viral infection to other astrocytes. Anti-HIV responses by astrocytes are much more potent in the cortex than in the cerebellum, therefore leading to a greater number of HIV+ cells in the cerebellum than in the cortex. (**C**) HIV+ macrophages and microglia, as well as astrocytes stimulate a powerful inflammatory response against infection (red dots), leading to damage and death of neurons (pink) and oligodendrocytes (green). (**D**) Circulating infected CD4+ T cells are trafficked into the brain and can infect glial cells via direct contact. (**E**) theorized route of infection from the meningeal lymphatic system, where HIV+ monocytes (blue) stimulate inflammation to permeabilize the vessels and carry infection into the CNS independent of the blood brain barrier. Created with BioRender.com.

**Figure 2 vaccines-09-00867-f002:**
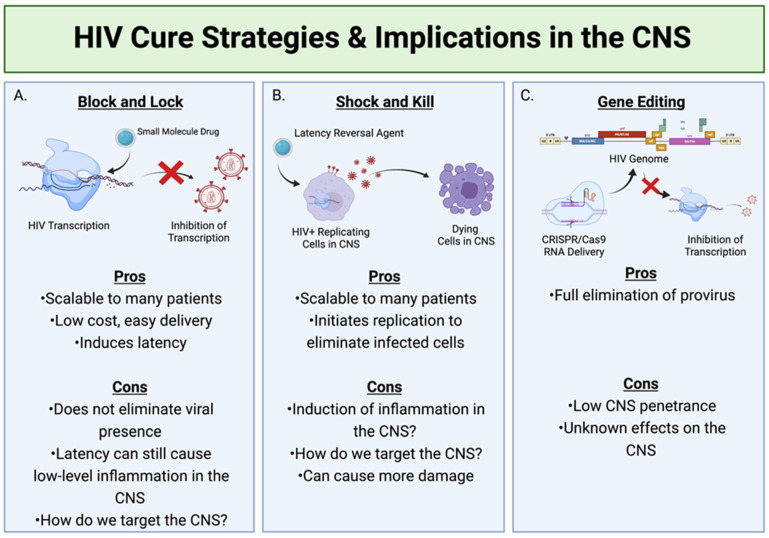
HIV cure strategies and their implications within the CNS. (**A**) Block and Lock involves the use of small molecule drugs to inhibit the transcription of the HIV genome, therefore stopping the formation of any new viral particles. (**B**) Shock and Kill utilizes latency reversal agents to activate HIV replication in latent cells, therefore making them targets for clearance by the immune system. (**C**) The utilization of CRISPR can deliver target RNA to the genome of infected cells harboring proviruses and eliminate these proviruses. Created with BioRender.com.

## Data Availability

Not Applicable.

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
