# Peer review of "HIV in the Brain: Identifying Viral Reservoirs and Addressing the Challenges of an HIV Cure"

_vaccines, 2021, doi:10.3390/vaccines9080867_

Round 1

Reviewer 1 Report

In the current review by Ash et al., the authors have discussed about the prominent HIV-1 reservoirs in the brain and the possible theories on they are developed in early stages of infection. The authors have also discussed some big questions remaining to be addressed as well as challenges in the path of HIV cure owing to the latent reservoirs. Some light is also shed on the technical challenges and identification of the reservoirs and the recent progress in this direction (This portion can become a separate section though). Overall, the review is short but precise and informative.

Some more comments are below:-

  • The lines in the manuscript are not numbered which makes it very difficult for a reviwer to pin-point corrections, should have been taken care of.

  • Fig1 needs improvement. This is the only figure in the review, and it is not self-explanatory. The reader has to delve into the legend to understand it. It would be worth labeling the cells at least. The authors could also be creative showing the cells within the brain diagram (optional) for events occurring in the brain compartment. Arrows could be used to depict the cyclic events/transfer of infected cells from peripheral regions of the body. Such changes would make the figure more informative even at quick glance.

  • In the section “Outstanding questions……”----Line-“Chinese rhesus macaques will exhibit shower…..”- I think you were supposed to write “slower” instead of shower. Please correct this typo. Hence, more careful editing of the draft would be worth doing.

  • The font style of the text in the last paragraph on page 6 is different from rest of the manuscript. Please correct it.

  • Consider adding at least one more figure describing the HIV cure strategies.

Reviewer 2 Report

The review ""HIV in the brain"" is very well written and is addressing the many challenges of a potential HIV cure that so many people are looking for.

Minor comments: sometimes the text is in a different format.

In Conclusion: although 2 persons are cured from HIV, are they still alive?

Author Response

The review ""HIV in the brain"" is very well written and is addressing the many challenges of a potential HIV cure that so many people are looking for.

Minor comments: sometimes the text is in a different format.

The whole document has had font normalized to Arial 11 point, with 1.5 spacing and justified margins.

In Conclusion: although 2 persons are cured from HIV, are they still alive?

We have included the most recent deceased status of the Berlin patient, revised in lines 300-301.

Reviewer 3 Report

In the proposed paper, “HIV in the Brain: Identifying Viral Reservoirs and Addressing the Challenges of an HIV Cure” the authors have addressed the paradigm of HIV reservoirs in the CNS and the relevant cell types, including astrocytes and microglia, that have been shown to harbor viral infection even with antiretroviral treatment. In particular, they focused on developmental aspects of astrocytes and microglia that lead to their susceptibility to infection, and how HIV infection propagates among these cells. They also addressed challenges of measuring the HIV latent reservoir, advances in viral detection assays, and how curative strategies have evolved in regard to the CNS reservoir. I believe that the results of this paper are interesting and helpful for biomedical scientists for treatment of HIV infection. I recommend this paper for publication in this journal, but after some corrections. My suggestions are given as follows:

  1. The abstract should be improved and extended so that it can reflect the overall content of the paper. Results and conclusions should be included as well.
  2. The introduction part should be elaborated to have a standard literature review. The reference list should be updated. I suggest the following recent works to the authors:
  • The role of prostitution on HIV transmission with memory: A modeling approach, Alexandria Engineering Journal 59(4):2513-2531,2020.
  • Modeling the mechanics of viral kinetics under immune control during primary infection of HIV-1 with treatment in fractional order. Physica A 2020; 545: 123816.
  • Global dynamics of a fractional order model for the transmission of HIV epidemic with optimal control. Chaos Solitons Fractals 2020; 138: 109826.
  • Chaotic dynamics of a fractional order HIV-1 model involving AIDS-related cancer cells. Chaos Solitons & Fractals 140:110272, 2020.
  1. The practical application of the obtained results in biology shall be stated in the conclusion section.

The results are correct. I recommend it for publication after the above-suggested revisions.

Author Response

Reviewer 3(response in red)

Comments and Suggestions for Authors

In the proposed paper, “HIV in the Brain: Identifying Viral Reservoirs and Addressing the Challenges of an HIV Cure” the authors have addressed the paradigm of HIV reservoirs in the CNS and the relevant cell types, including astrocytes and microglia, that have been shown to harbor viral infection even with antiretroviral treatment. In particular, they focused on developmental aspects of astrocytes and microglia that lead to their susceptibility to infection, and how HIV infection propagates among these cells. They also addressed challenges of measuring the HIV latent reservoir, advances in viral detection assays, and how curative strategies have evolved in regard to the CNS reservoir. I believe that the results of this paper are interesting and helpful for biomedical scientists for treatment of HIV infection. I recommend this paper for publication in this journal, but after some corrections. My suggestions are given as follows:

  1. The abstract should be improved and extended so that it can reflect the overall content of the paper. Results and conclusions should be included as well.

We have lengthened the abstract to include the addition of curative strategies and implications for the field and in the clinic.

  1. The introduction part should be elaborated to have a standard literature review. The reference list should be updated. I suggest the following recent works to the authors:
  • The role of prostitution on HIV transmission with memory: A modeling approach, Alexandria Engineering Journal 59(4):2513-2531,2020.
  • Modeling the mechanics of viral kinetics under immune control during primary infection of HIV-1 with treatment in fractional order. Physica A 2020; 545: 123816.
  • Global dynamics of a fractional order model for the transmission of HIV epidemic with optimal control. Chaos Solitons Fractals 2020; 138: 109826.
  • Chaotic dynamics of a fractional order HIV-1 model involving AIDS-related cancer cells. Chaos Solitons & Fractals 140:110272, 2020.

Thank you for the reference suggestions, we have reviewed them and they do not apply to our topic, so we have not included them in the revisions.

  1. The practical application of the obtained results in biology shall be stated in the conclusion section.

We have included an addition of application of results addressed in lines 306-310.

The results are correct. I recommend it for publication after the above-suggested revisions.